# Automated Method Design for Cancer Image Classification by Differential Evolution and Ensembling

Natalia Oviedo Acosta[1,2][0009−0002−9477−0454], Stefan Klein[1][0000−0003−4449−6784], and Martijn P.A. Starmans[1,2][0000−0001−5086−7153]

[1] Department of Radiology and Nuclear Medicine, Erasmus MC, Rotterdam, the Netherlands
{n.oviedoacosta, m.starmans, s.klein}@erasmusmc.nl
[2] Department of Pathology, Erasmus MC, Rotterdam, the Netherlands

**Abstract.** Developing deep learning models for cancer image classification requires many method design choices, such as in data preprocessing, model architecture, hyperparameters and training procedures. Typically, these are manually tuned, a process that is time-consuming, expert-dependent, and often irreproducible. To address these challenges, we propose an automated machine learning (AutoML) framework that optimizes model design without human intervention. To ensure a comprehensive exploration of diverse architectures and hyperparameter configurations, we define a search space based on state-of-the-art literature in cancer imaging. Our framework employs Differential Evolution and Hyperband (DEHB), which integrates evolutionary search algorithms to balance search space exploration and exploitation, combined with adaptive resource allocation to mitigate the high computational cost of training multiple models. To enhance model robustness and reduce overfitting in data-limited scenarios, we incorporate ensembling. We validate our approach on four public cancer classification datasets encompassing 750 patients with either MRI or CT. The proposed framework demonstrates higher performance when compared to a DenseNet-121 baseline. While exploring multiple configurations, our approach reduces training time by a factor of two to five compared to the baseline. By automating model design and improving generalization across datasets, our framework has substantial potential for broad applications across cancer imaging, thereby streamlining deep learning model development.

**Keywords:** AutoML · evolutionary algorithms · hyperparameter optimization · computer-aided diagnosis

## 1 Introduction

Deep learning has substantially advanced cancer image analysis by enabling precise and early diagnosis through AI-driven models [17]. However, developing these models remains a labor-intensive process that requires numerous decisions

regarding data preprocessing, model architecture, hyperparameters, and model training. Commonly the majority of these choices are made manually through a heuristic trial-and-error process, which is time-consuming, prone to overfitting [10], limits reproducibility [12,21], may lead to suboptimal solutions [12] and can be resource-intensive, making it inefficient [24,16].

Among cancer imaging tasks, classification poses unique challenges, as it often relies on a limited number of image-level labels, leading to small and heterogeneous datasets that make model development highly sensitive to hyperparameter choices and prone to overfitting, where models may learn spurious correlations instead of generalizable patterns [25]. To address these challenges, we propose an automated method design framework based on Automated Machine Learning (AutoML) [12] to streamline and optimize deep learning model development for cancer image classification. AutoML has been gaining significant traction in recent years and is increasingly being adopted in medical imaging. Promising results have been reported in both segmentation [4] and classification tasks [7], with several recent studies providing comprehensive reviews and benchmarks tailored to clinical applications [14,2]. As a next step in this direction, our framework automates key design decisions, including data preprocessing, model selection, hyperparameter tuning, and model training, thereby eliminating manual intervention. We leverage Differential Evolution and Hyperband (DEHB) [3], a state-of-the-art AutoML optimization algorithm, to efficiently explore large hyperparameter spaces and improve both convergence speed and model performance. We hypothesize that integrating DEHB into the model development pipeline will substantially accelerate training compared to manual tuning. Additionally, we incorporate ensembling strategies to regularize DEHB optimization and improve overall performance.

## 2  Methods

To automate model design, given a training dataset $D_{\text{train}}$ and a validation dataset $D_{\text{val}}$, we define an optimization function with the objective of identifying the optimal hyperparameter configuration $x^*$ that minimizes the average validation loss $\mathcal{L}$ over $K_{\text{training}}$ iterations (e.g., cross-validation). We formulate the optimization of any method design choice, including data preprocessing, model architecture, and model training, as a combined method selection and hyperparameter optimization problem. Formally, we express this as follows:

$$x^* = \arg\min_{x \in X} \frac{1}{K_{\text{training}}} \sum_{k=1}^{K_{\text{training}}} \mathcal{L}(x, D_{\text{train},k}, D_{\text{val},k}, b), \tag{1}$$

where $x$ represents a hyperparameter configuration, $X$ denotes the hyperparameter search space, and a given computational budget $b$ (i.e., number of epochs).

Manual tuning and conventional tuning strategies such as exhaustive grid search or random search over $X$ are computationally expensive, as these methods lack efficient resource allocation mechanisms. To address this, we propose

to employ DEHB, which effectively balances the exploration of the search space with a focus on promising solutions through an evolutionary algorithm, while efficiently managing computational resources using Hyperband [16]. By leveraging Differential Evolution (DE) [18] for guided search and Hyperband for adaptive resource allocation, DEHB provides a more efficient and scalable approach to hyperparameter optimization, making it particularly well-suited for large and complex search spaces as well as resource-intensive model training.

### 2.1   Differential Evolution and Hyperband (DEHB)

As presented in Algorithm 1, DEHB is initialized by evaluating a population of $Q$ candidates, i.e., set configurations of hyperparameters randomly sampled from $X$. Initially, each configuration is trained on the minimum budget $b_{\min}$, and its validation loss $\mathcal{L}(x, D_{\text{train}}, D_{\text{val}}, b_{\min})$ is measured. After initialization, Hyperband dynamically allocates computational resources by discarding poorly performing configurations through successive halving. Configurations are evaluated in progressive stages, where in each stage only the top $1/\eta$ fraction of candidates continues to the next stage with an increased budget $b$, while the rest are eliminated. The selection is based solely on the validation loss observed at the current budget level, without the use of any patience mechanism. Parameter $\eta$, known as the aggressiveness factor, controls how many configurations are discarded. The number of remaining candidates at stage $s$ is therefore $Q \cdot (1/\eta)^s$, making the reduction schedule deterministic. The budget $b$ increases at each stage according to the Hyperband schedule, typically growing multiplicatively with $\eta$. This process iterates until a predefined $b_{\max}$ is reached or only one configuration remains. Once Hyperband has selected the top-performing candidates for further evaluation, DEHB extends this process by continuing the search with evolutionary steps. The general procedure stops once all $P$ configurations have been evaluated and the $x^*$ is selected based on the validation loss.

**Mutation** Each configuration $x_i$ represents a vector of chosen hyperparameters. New candidate configurations, or mutant vectors $v_i$, are generated through a mutation process that introduces controlled variations based on existing ones. Specifically, mutation is performed as follows:

$$v_i = x_r + F \cdot (x_a - x_b),$$

where $x_r, x_a, x_b$ are randomly selected hyperparameter configurations from the population, and $F$ is a scaling factor controlling the magnitude of change within the range (0,1].

**Crossover** After mutation, a new candidate vector $u_i$ is generated through crossover, where each parameter is inherited from the original vector $x_i$ or the mutant vector $v_i$. Each configuration vector (e.g., $x_i$) consists of $N$ hyperparameters, denoted as $x_i^j$, $v_i^j$, and $u_i^j$, where $j \in \{1, 2, \dots, N\}$ indexes the individual

---

**Algorithm 1** Differential Evolution and Hyperband (DEHB)

---

**Require:** Search space $\mathcal{X}$, budgets $b_{\min}$, $b_{\max}$, reduction factor $\eta$, max evaluations $P$
 1: Randomly sample $Q$ configurations from $\mathcal{X}$ into $P_{b_{\min}}$
 2: Evaluate all configurations in $P_{b_{\min}}$ at budget $b_{\min}$
 3: **while** total function evaluations $< P$ **do**
 4:    **for** each budget $b$ in $\{b_{\min}, \eta b_{\min}, \eta^2 b_{\min}, \ldots, b_{\max}\}$ **do**
 5:       Select top $1/\eta$ fraction of configurations from previous budget as parent pool
 6:       **for** each configuration $x_i$ in the parent pool **do**
 7:          Generate mutant vector $v_i$
 8:          Generate trial vector $u_i$ via crossover between $x_i$ and $v_i$
 9:          Evaluate $u_i$ on budget $b$
10:          Replace $x_i$ in the population if $u_i$ performs better
11:       **end for**
12:    **end for**
13: **end while**
14: **return** $x^*$: best configuration across all budgets

---

hyperparameters. Crossover is performed independently for each hyperparameter $j$, selecting between $v_i^j$ and $x_i^j$ according to the following rule:

$$u_i^j = \begin{cases} v_i^j, & \text{if } (z \sim \mathcal{U}(0,1)) \leq Cr \text{ or } (j = j_{\text{rand}}) \\ x_i^j, & \text{otherwise} \end{cases}$$

where $Cr$ is the crossover probability, determining the likelihood of inheriting parameters from $v_i$. The index $j_{\text{rand}}$, randomly selected from $\{1, 2, \ldots, N\}$, ensures that at least one parameter comes from $v_i$, to prevent $u_i = x_i$. This process allows crossover to integrate information from the mutant vector while preserving useful characteristics from the original configuration.

**Selection** The selection process then compares the performance of $u_i$ against its corresponding parent configuration $x_i$ using the optimization function in Equation 1. If $u_i$ achieves a lower validation loss than $x_i$, it replaces $x_i$ in the population for the next iteration. Otherwise, $x_i$ is retained.

### 2.2   Search space

The performance of an AutoML framework highly depends on the search space, i.e., the potential hyperparameter configurations. Since non-architectural design choices such as data preprocessing, augmentation, model selection, and model training can be equally important as architectural ones [13,6], we construct a comprehensive search space covering all these aspects, including discrete and categorical hyperparameters, which motivates our choice of DEHB as the optimization strategy. To reduce computational burden, we set a GPU memory constraint to filter out infeasible configurations and maintain efficient training times. The complete search space is illustrated in Figure 1.

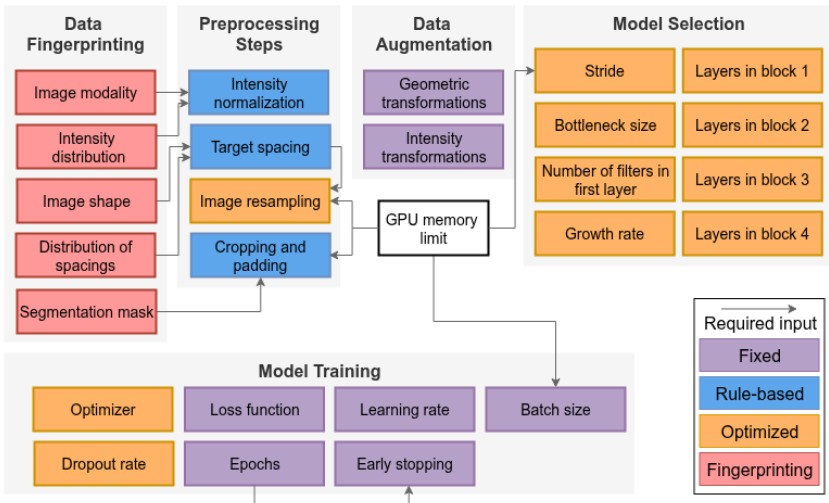

**Fig. 1.** Overview of the proposed search space for cancer image classification. Default values are shown in bold.

Inspired by nnU-Net [13], we introduce dataset fingerprinting to extract key information about the input data, including modality, intensity distribution, image shape, and spacing. These properties are used to define rule-based constraints that both filter infeasible configurations and determine default hyperparameter values within our AutoML search space.

Preprocessing includes intensity normalization and resampling. To account for differences in resolution due to scanner variations, we included different target spacings. For cropping and padding, we constrained the image sizes to a maximum of 512 voxels in all three dimensions. As our datasets contain the tumor segmentation masks, we applied tumor-centered cropping to focus on the region of interest while reducing unnecessary background. To prevent test-time bias, we applied all preprocessing steps described above exclusively to the training data.

The backbone architecture used is DenseNet [11], which has demonstrated strong performance in cancer imaging tasks due to its efficient feature propagation and parameter efficiency, making it suitable for learning from limited and heterogeneous datasets [26]. Search ranges for bottleneck size, growth rate, initial filter count, and block depth were defined based on default DenseNet-121 values [11] and expanded to explore broader architectural variations.

For training parameters, the loss function was fixed to cross-entropy loss, a standard choice for classification tasks as stated in [19]. The learning rate is fixed, as prior work has shown that optimizing it can be unreliable under multi-fidelity optimization frameworks like Hyperband [15]. To ensure stable optimization with a batch size of one, gradient accumulation was combined with instance normalization to maintain consistent updates and training dynamics.

### 2.3   Ensembling

While DEHB aims to identify a single optimal configuration, the optimization landscape in medical imaging—particularly with small, heterogeneous datasets—can be noisy [17,8], and AutoML methods such as DEHB are prone to overfitting on the validation dataset [1]. To mitigate this and improve generalization, we construct ensembles by averaging predictions from multiple models trained with diverse configurations selected by DEHB, ensuring that only competitive candidates contribute while maintaining diversity across method design choices. Given a set of $M$ selected hyperparameter configurations $\{x_1, \ldots, x_M\}$, each of which is evaluated across $K$ different data splits (e.g., cross-validation folds), we obtain a total of $M \cdot K$ trained models.

The ensemble prediction $\hat{y}$ for an input $z$ is computed as

$$\hat{y} = \frac{1}{M \cdot K} \sum_{m=1}^{M} \sum_{k=1}^{K} \hat{y}_{m,k}(z),$$

where $\hat{y}_{m,k}(z)$ denotes the prediction made by the model trained with configuration $x_m$ on fold $k$. This formulation allows for flexibility in ensemble design: setting $M = 1$ yields an ensemble over data splits, while increasing $M$ incorporates configuration-level diversity.

# 3 Experiments and results

## 3.1 Dataset

For this study, four datasets from the publicly available WORC Database [23] are used, comprising 750 anonymized patients with various tumor types. Each dataset includes an MRI or CT scan, a tumor segmentation, and a tumor type label serving as the prediction target. The datasets contain 115 patients with liposarcoma or lipoma (Lipo), 203 with desmoid-type fibromatosis or extremity soft-tissue sarcomas (Desmoid), 186 with primary solid liver tumors (Liver), and 246 with gastrointestinal stromal tumors or similar intra-abdominal tumors (GIST).

## 3.2 Experimental Setup

To evaluate the framework, two baselines were used: (1) the WORC framework [22], previously validated on these datasets, and (2) DenseNet-121 [11] without hyperparameter tuning, trained from scratch for 100 epochs with 5-fold cross-validation using default settings.

For DEHB, the maximum and minimum fidelity levels (number of epochs) were set to $b_{\max} = 50$ and $b_{\min} = 30$, respectively. Other parameters followed default values [3]: $\eta = 2$, $F = 0.5$. Each dataset was split into 80% training and 20% test sets. All model selection and cross-validation procedures, including those for DEHB and DenseNet-121, were conducted within the training set using 5-fold cross-validation. DEHB performed $P = 10$ function evaluations per fold, selecting one best configuration per fold. These five configurations were retrained and evaluated across all five folds to obtain robust performance estimates, yielding 25 trained models per dataset. All experiments were run on $8\times$NVIDIA A40 48 GB GPUs using MONAI [5] for efficient medical image processing.

Model performance was primarily measured using ROC-AUC [9]. DEHB was evaluated in three ways: (1) the best configuration (lowest average validation loss across five folds) was retrained on the full 80% training set and tested on the 20% holdout; (2) a 5-Model Ensemble (5ME) was created by training the top configuration across all five folds ($M = 1$, $K = 5$); and (3) a 25-Model Ensemble (25ME), where the top five configurations were retrained across five folds ($M = 5$, $K = 5$), and their predictions averaged.

**Computation time** Wall-clock time was measured for each dataset to assess the efficiency of our framework, using $P = 50$ configurations. For the DenseNet-121 baseline, training was performed with default settings using a single configuration for 100 epochs. Since training all $P$ configurations individually for the baseline would be computationally intensive, we estimated the cost by training once on the larger datasets (Desmoid and GIST) and averaging five runs on the smaller ones (Lipo and Liver). This simulation was intended to approximate the total wall-clock time that would be required if 50 different configurations, equivalent to those evaluated by DEHB, were trained from scratch using the DenseNet-121 baseline, enabling a fair comparison of computational cost.

### 3.3   Results

Table 1 compares the performance of different methods across all four datasets. The DEHB framework alone slightly underperforms both baselines on most datasets. However, incorporating the 5ME strategy with DEHB improves performance across most datasets, with a higher performance increase observed with DEHB + 25ME in all datasets.

**Table 1.** ROC-AUC for the four datasets, evaluated under three different setups: (1) WORC in $100\times$ random-split cross-validation, (2) DenseNet-121 in 5-fold cross-validation, and (3) DEHB using a fixed 80/20 train/test split. The second row indicates the number of samples ($N_s$) for each dataset.

| Model | Lipo | Liver | Desmoid | GIST |
|---|---|---|---|---|
| $N_s$ | 115 | 186 | 203 | 246 |
| *Baselines* | | | | |
| WORC | 0.83 | 0.80 | 0.82 | 0.77 |
| DenseNet-121 | 0.82 | **0.88** | 0.80 | 0.61 |
| *Framework* | | | | |
| DEHB | 0.78 | 0.83 | 0.73 | 0.73 |
| DEHB + 5ME | 0.83 | 0.86 | 0.80 | 0.78 |
| DEHB + 25ME | **0.87** | 0.87 | **0.83** | **0.81** |

**Computation time** Our framework showed substantially lower computational cost across all datasets compared to the simulated 50x DenseNet-121 baseline. On Lipo, DEHB completed all $P = 50$ configurations in 95 hours versus 650 for the baseline. For Liver, DEHB required 220 hours (vs. 900), and for Desmoid, 340 hours (vs. 1100). On GIST, the largest dataset, DEHB finished in 216 hours (9 days), while the baseline was projected to take 3600 hours (150 days). These results demonstrate a 2–5x speedup, indicating that the framework scales efficiently and enables automated model design without compromising runtime feasibility.

## 4   Discussion and conclusion

This study presents an AutoML framework for cancer image classification that combines DEHB with ensembling for automated method design. DEHB identified multiple high-performing models with similar validation performance, as observed in five-fold cross-validation. Instead of relying solely on a single configuration, we aggregate these diverse models through ensembling to enhance robustness and generalization. Ensembling helps counteract overfitting by averaging individual model biases. Although the improvement varies across datasets, our results show that ensembles consistently match or outperform both the best individual DEHB model and the baselines in the majority of cases, supporting

their complementary role. Moreover, DEHB substantially reduced computational time compared to the baseline by leveraging adaptive resource allocation, avoiding exhaustive training and minimizing the need for manual tuning. Further efficiency gains may be possible through techniques like reduced precision computations [20]. While our evaluation offers a practical comparison basis, the fixed 80/20 train/test split used for DEHB may limit comparability with baselines trained under different protocols. Future work will focus on aligning evaluation strategies. Additionally, although DenseNet-121 was used as the backbone, extending the framework to other architectures could offer broader insight into DEHB's generalizability.

Our results highlight the effectiveness of combining DEHB with ensembling to achieve competitive performance while significantly reducing computational cost. By efficiently exploring configurations across preprocessing, architecture, and training, the framework enables fully automated model design. This not only accelerates development but also improves reproducibility, reduces overfitting, and avoids suboptimal choices. These strengths make DEHB a compelling solution for optimizing deep learning in cancer imaging, especially in resource-constrained settings, and support more scalable, reproducible AI-driven diagnostics.

**Acknowledgments.** This work is part of the AIID project, funded by the Dutch Research Council (NWO) under the AiNed Fellowship Grants programme (project number NGF.1607.22.025).

**Disclosure of Interests.** The authors have no competing interests to declare that are relevant to the content of this article.

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
