# OpenReview forum: "Automated method design for cancer image classification by Differential Evolution and Ensembling"
_MICCAI.org/2025/Workshop/MSB_EMERGE — MSB EMERGE 2025 Oral_

### Official Review · Reviewer_nRPi · 2025-07-07

**Recommendation:** 4
**Confidence:** 4

**Clarity:**

The paper is clear and well-written, with minor areas for improvement in clarity

**Feedback:**

Except for the listed weaknesses:

-   It is not clear why the learning rate is not included in the search space
-   Hyperband should be cited when mentioned first, page 2 bottom
-   Would be interesting how different settings influence performance.
    Do the best performing models only vary in single parameters or can they be very different?
-   The method could be extended beyond cancer applications.

**Justification:**

The paper is well written, however, there are some open questions that are not clear, i.e. model merging for baselines and computation time calculation.

**Reproducibility:**

Sufficient amount of details available for reproducing the main results, and open access is provided (or promised upon acceptance) to source code and/or data

**Strengths:**

1.  The paper is well written and easy to follow.
2.  Designing a automated machine learning framework for cancer image classification
    depicts a valuable task, as hyperparameter optimization for such small and heterogeneous datasets
    is cumbersome.
3.  The authors compare their method against two baseline models, showing slight improvements.

**Summary:**

The authors test the ability of the Differential Evolution and Hyperband (DEHB) algorithm to construct an automated machine learning framework for cancer image classification. To do so, they define a search space for identification of optimal hyperparameter settings and evaluate the effectiveness of combining different models using ensembling. The method is trained on four different classification task of the WORC dataset and compared to two baseline methods.

**Weaknesses:**

4.  It should made clear that section 2.1 is a background section, with no novelty coming
    from the authors.
5.  In section 2.2 the authors write that their data fingerprint is "inspired" by nnU-Net.
    However, no further clarification about the fingerprint is provided.
    Is it the same fingerprint as for nnU-Net? Then the authors should say so and not use "inspired".
    Otherwise, further clarification is needed.
6.  Training of the baseline models is not completely clear. The authors write that they
    use 5-fold cross-validation for DenseNet. How are the different models used? Are they
    merged following 2.3 or is only the best model used? Is the WORC algorithm retrained
    for the data split in this paper? If so, which numbers for random search iterations N_{RS}
    and ensemble size N_{ENS} are used?
7.  The comparison for the computation time is not clear. Authors write that DenseNet was
    only trained with default settings but also that the total cost was calculated for P=50.
    If it was only trained with default settings, the computation time should only be computed
    for one setting, or five taking the cross-validation into account.

---

### Official Review · Reviewer_ehNw · 2025-07-08

**Recommendation:** 3
**Confidence:** 4

**Clarity:**

The paper is clear and well-written, with minor areas for improvement in clarity

**Feedback:**

Please address the weaknesses above, especially contextualizing the proposed method w.r.t. the existing literature. Please also proofread the paper. I found at least 1 grammatical error: (page 2) "image-level labels. Resulting" should be "image-level labels, resulting".

**Justification:**

While the paper is generally well written, there are several issues with the paper, including the lack of references to relevant literature for both design choices and experimental comparisons.

**Reproducibility:**

Sufficient amount of details available for reproducing the main results, and open access is provided (or promised upon acceptance) to source code and/or data

**Strengths:**

1. The paper is well written for most parts (see Weaknesses), and Section 2.1 especially is detailed is easy to follow.
2. The experimental setup is quite extensive with cross-validation and a large search space (Fig. 1).

**Summary:**

Evolutionary AutoML + ensembling for medical image classification

**Weaknesses:**

1. The authors need better references to support some of their claims and in general literature review.
(a) The authors have cited only a single prior work ([3]) for AutoML, which is surprising because there are much more relevant and recent surveys that the authors can cite when talking about AutoML in general [A] and for medical imaging [B] in particular. Similarly, there is at least one (very) popular method [C] for neural evolution-based AutoML that is completely relevant and should at least be discussed, if not compared against.
(b) DenseNet is chosen because it "has demonstrated strong performance in cancer imaging tasks due to its efficient feature propagation and parameter efficiency, making it suitable for learning from limited and heterogeneous datasets." This is quite a sweeping yet generic statement that is not supported by any references, and therefore it is hard to agree.
(c) Similarly, the authors fix the loss function and the learning rate, but the rationale for doing so is not supported by the associated references [14, 15], which are a deep learning textbook and an LR-scheduler paper.

2. It's not quite clear why the loss function and the learning rate were not also included in the search space.

3. The authors perform a tumor-location-dependent cropping to obtain a region of interest (ROI). However, if this is done for the test images too, doesn't this mean that this prediction model needs access to test sets' tumor segmentation masks? How is this a fair and held-out test then?

4. For DEHB, the authors define the computational budget $b$ in terms of the number of training epochs. While the number of epochs provides a rough sense of training duration, it does not accurately represent computational budget across models of differing complexity. A more meaningful and fair comparison could be achieved by measuring the budget in terms of wall-clock time, floating point operations (FLOPs), MACs (multiply-and-accumulate operations), or total GPU hours.

5. Unclear if the DenseNet-121is trained from scratch or based on the ImageNet-pretrained weights.

6. Unclear motivation for using a mixed-modality database ("Each dataset includes an MRI or CT scan ..."). It makes it quite hard to assess if one DEHB works better for one modality or both.

7. The computational resources required (8*48 GB of GPU VRAM, several days) considerably weaken the authors' argument about the purported efficiency of their method.

8. The authors conduct 5-fold cross validation but do not report any measure of variance (e.g., std. dev., variance) in Table 1.

9. What does this sentence mean? "Among cancer imaging tasks, ... image-level labels." This is incorrect, since any other medical imaging task that involves dense prediction (e.g., localization, detection, segmentation, etc.) pose more "unique challenges".

[A] https://doi.org/10.1007/s10115-023-01935-1
[B] https://doi.org/10.1016/j.compmedimag.2024.102441
[C] https://doi.org/10.1145/3321707.3321721

---

### Official Review · Reviewer_hqL2 · 2025-07-09

**Recommendation:** 4
**Confidence:** 3

**Clarity:**

The paper is clear and well-written, with minor areas for improvement in clarity

**Feedback:**

**Suggestions:**

I recommend that, in addition to eliminating infeasible configurations (as discussed in Section 2.2), the authors also consider proposing initial configurations that could yield better performance. For instance, starting with over-parameterization and an increased number of epochs may be beneficial. This approach could help reduce the overall time consumed.

Ablating the common loss functions, including the learning rate for training, would benefit their contribution.

The authors could provide information on the choices that were successful or the best selections for each dataset. This can also be included in Figure 1.

The number of dataset samples can also be included in Table 1. Additionally, the weaknesses section also includes suggestions.

**Justification:**

Overall, the paper is useful for practical scenarios, and evaluated on 4 datasets. Moreover, the improvements thatthe proposal offers have been clearly stated with explainations.

**Reproducibility:**

Sufficient amount of details available for reproducing the main results, and open access is provided (or promised upon acceptance) to source code and/or data

**Strengths:**

- The authors provide clear descriptions of the training configurations.
- Training on small datasets often involves numerous hyperparameters, which can lead to convergence issues. Unlike grid search, which is resource-intensive, the authors suggest using Differential Evolution Hyperband (DEHB) to generate training configurations.
- It's beneficial to note that several training configurations from the search space 'X' have been eliminated for their specific applications.
- The proposal has been evaluated on four datasets, which aids in analyzing the results.
- The algorithm adds significant value to the paper's explanation, and the accompanying figure is comprehensive.

**Summary:**

Deep learning for medical imaging enounters scarce datasets for training. As a result, training is challenging due to overfitting on the training samples, resource-intensive in some cases where it's hard to train, and leads to suboptimal solutions. Additionally, the choice of hyperparameters is crucial for such datasets. To mitigate this, the authors propose a design framework based on AutoML for cancer classification.  They leverage Differential evolution hyperband (DEHB) to explore the large hyperparameter choice, thereby improving the convergence speed and model performance. Next, they use ensemble starergies to regularize DEHB optimization and improve performance. The authors validate the effectiveness of their proposal on four datasets and demonstrate that their method performs competitively compared to the baselines. Additionally, the authors provide detailed computational time for their method, which highlights their improvement.

**Weaknesses:**

- The proposal outlined in the methodology section of the paper is not clear. In Section 2.1, the overall terminology of Hyperband is explained in a generic manner. I would like to see how this applies specifically to the cancer classification task. The authors could start directly with notations for the dataset and keep the text specific rather than generic.

- Additionally, the methods section should have commenced with notations relevant to the application task at hand, namely, the cancer classification task. Characteristics of the dataset, such as the number of samples available for training and any class imbalance (if present), should have been introduced. As it stands, it is too generic.

- What is the upper bound of ‘X’ for Hyperband [6]?

- How does Hyperband discard poorly performing configurations? Is this done solely based on validation loss? What is the patience level applied in this context?

- When the goal is to find the optimal x*, which may lead to the best-performing model, why is ensembling necessary? Doesn’t this contradict the objective of obtaining x*?

- Why is ensembling necessary after obtaining good models with generated configurations?
If the goal is to create ensembles, multiple models can be trained with random configurations, followed by an ensemble of the models’ outputs. Wouldn't this also lead to similar performance (assuming random configurations converge)

- Related to this, why is the ensemble effective in certain cases, particularly for Table 1 (Lipo)? Despite having 25 ensembles, the improvement in model performance is not significant. I disagree with the claim that “When ensembled, these models showed improved generalization, suggesting they overfit to different aspects of the training data,” as the ensemble version does not consistently outperform the baseline significantly, according to Table 1.

- I recommend including error margins for Table 1, especially for the ensembles and Differential Evolution Hyperband (DEHB).

- The baseline consumes 650 hours of computation time. Could the authors provide more details about this? Is this duration due to the elimination of grid search options?